# Geoethics for Nudging Human Practices in Times of Pandemics

**Eduardo Marone** [1,2,*] **and Martin Bohle** [2,3] 

1   Centre for Marine Studies (CEM), International Ocean Institute Training Center for Latin America and the Caribbean (IOITCLAC), Federal University of Paraná (UFPR), Pontal do Paraná 83255-976, Brazil

2   International Association for Promoting Geoethics (IAPG), 00143 Rome, Italy; martin.bohle@ronininstitute.org

3   Ronin Institute for Independent Scholarship, Montclair, NJ 07043, USA

*   Correspondence: edmarone@gmail.com

**Abstract:** Geoscientists developed geoethics, an intra-disciplinary field of applied philosophical studies, during the last decade. Reaching beyond the sphere of professional geosciences, it led to professional, cultural, and philosophical approaches to handle the social-ecological structures of our planet 'wherever human activities interact with the Earth system'. Against the backdrop of the COVID-19 and considering geoscientists' experiences dealing with disasters (related to hazards like tsunamis, floods, climate changes), this essay (1) explores the geoethical approach, (2) re-casts geoethics within western philosophical systems, such as the Kantian imperatives, Kohlberg scale of moral adequacy, Jonas' imperative of responsibility, and (3) advances a 'geoethical thesis'. The latter takes the form of a hypothesis of a much broader scope of geoethics than initially envisioned. That hypothesis appears by suspecting a relationship between the relative successes in the COVID-19 battle with the positioning of agents (individual, collective, institutional) into ethical frameworks. The turmoil caused by the COVID-19 pandemic calls for the transfer of experiences between different disciplinary domains to further sustainable governance, hence generalizing the geoethical approach. It is emphasized that only when behaving as responsible and knowledgeable citizens then people of any trade (including [geo-]scientists) can transgress the boundaries of ordinary governance practices with legitimacy.

**Keywords:** geoethics; social-ecological systems; ethical imperatives; COVID-19 pandemic; responsible science

## 1. Introduction

*"We know what is at stake only when we know that it is at Stake"*. [1] (p.88)

The concept of geoethics did emerge as a deontological development [2]. However, the scholars who developed geoethics implicitly explored questions of governance. Subsequently, it is assumed that better governance results are obtained when higher standards of moral adequacy of human agents prevail in a society [3]. This essay explores that assumption in a broad context taking the COVID-19 pandemic as the reference frame [4]. In a first instance, geoethics does propose obligations (ethical norms such as the Geoethical Promise [5], Cape Town Statement on Geoethics [6], or Geoethical Rationale [7]) to guide the behaviors of an individual actor. Geoethics, when rethinking, for example, its agent-centricity in terms of institutional agents, is about appropriate governance practices.

The initial narrative of the development of geoethics is that of geologists establishing deontological codes for their professional dealings [2]. Professional geologists and other geoscientists operate in context-depending societal and natural settings that often involve significant social and economic

claims [8]. One of the leading and original contributions of geoethics was putting axiomatic emphasis on the virtues of the responsible professional agent who operates context-depending and knowledge-based [9]. Subsequently, studies in geoethics did a sketch of the 'geoethical approach', namely how human agents appropriately interact with the Earth system from the geosciences perspective, not just as Earth's natural scientists (Earth scientists, geoscientists) but also as citizens. Within ethics of human and natural systems (see for example [10,11]) geoethics is situated at the interface of environmental ethics, sustainability ethics, and professional ethics with the dedicated subject of the abiotic compartments of the Earth system [12].

In a further narrative, additional ontological perspectives and epistemological analysis were added to the initial deontological sketch of geoethics. In that context, the group of scholars, Earth scientists or geoscientists dedicated to developing of geoethics, initiated cooperation with colleagues from the areas of philosophy and the social sciences [13]. The initial concepts were expanded, updated, and reformulated [14]. In due course of exploring the implications of geoethical approach, the planet Earth stopped being seen overwhelmingly from the natural sciences; perspectives of culture, communication, and sense-making emerged as part of a more comprehensive picture of the human condition [15]. Likewise, anthropocentric visions of submitting nature to human practices, like ecomodernism, were eluded. In the same process of discoveries, the prowess of human practices to alter Earth profoundly is acknowledged; hence using the label 'Anthropocene' to characterize present geological and human times [16]. Subsequently, the dynamics of planet Earth were perceived in a more integrated manner, considering its non-linear system behaviors, its complex dynamics full of feedbacks, and irreversible development paths [17]. Concepts like the 'social-ecological systems' [18,19] were embraced and are broader, more umbrella-like; furthermore, the concept of complex-adaptive dynamics explains why system behaviors may appear 'wicked' to the observer [20].

Analyzed these narratives in hind-cast, the development of geoethics (or of the 'geoethical approach') begins with obligations that can be related to Kant's Categorical Imperative and the challenges posed by his three fundamental questions (what to know, to do, and to hope?) for any human agent [21]. However, as pointed by Max Weber, a Kantian structure deemed incomplete because lacking to consider the responsibility of the human agent [22,23]. Here, Jonas' imperative of responsibility gives an additional framework, namely, to ensure the preservation of genuinely human life, which is most paramount in pandemical times than ever [1,24]. To extend the sketch of obligations, Bunge's moral principle is added, namely, while presenting the right to happiness the duty reigns of helping genuinely human and other biological forms of life [25]. To complete the philosophical framework, Kohlberg's scale of moral adequacy provides an approach to gauge human behaviors and practices [26]. Higher ethical standards, sound science support, and good governance are vital aspects of the fight and recovery from the pandemic, in the same way as when mitigating natural hazards that usually is studied by geosciences. This essay describes how the evolution of the notion geoethics along this philosophical path opens an application space of geoethics beyond geosciences.

In times of writing, the expanding COVID-19 pandemic (from the Greek, pan = everything/all; demos = people), the emergent vision of the Earth system and the World as complex-adaptive social-ecological systems pose to all Earth scientists the duty to contribute with their scientific expertise [27]. Therefore, we suppose that some experiences that are quite advanced within geoethics can contribute to remedying the shock of the COVID-19 pandemic. A substantial share of geoscience expertise, including geoethics, is deployed to handle natural hazards, technological infrastructures, and human-induced disasters that relate to them; landslides, earthquakes, tsunamis, and floods may serve as examples. Suchlike geoscience experiences do not necessarily contribute to the treatment of the health issues itself. However, they may help with the adaptation, resilience, and reorganization of the (global) social-ecological systems after the initial strike of disasters, including, if suitably generalized, the aftermaths of a health pandemic.

Finally, and iterating the word of caution, one would be mistaken assuming detailed lessons from geoethics on topics, like specific handling of the COVID-19 health pandemic, apparently so distant

from fundamental subjects and practices of geosciences. Nevertheless, transdisciplinary contributions can be distilled from geoethics, which should help in post-pandemic times and when enduring them. Overall, the geoethical approach could also be valid in the management of the COVID-19 pandemic, such as to follow the best available scientific advice, stay human-agent centric, and to apply high ethical standards. Experiences in several fields [28] indicate that people acting geoethics-like as citizens, jointly and collaboratively, may move beyond the regular governance practices.

The following sections of the essay describe (Section 2) the geoethical approach (in short 'geoethics'), situate geoscience expertise and geoethics in times of pandemics (Section 3), and terminate the essay with a call why not to fear 'The New', 'The Unknown', and 'The Counter-Intuitive' that, metaphorically speaking, label the present.

## 2. Materials and Methods

Building on the introduction, this section describes, in the form of additional materials and methods, how geoethics relates to a pandemic, why this relationship is an opportunity to size with caution, and why some re-design of geoethics deems appropriate.

### 2.1. Geoethics Situated in Pandemics

For the public, a pandemic is mostly limited to a health problem. This understanding of the concept of a pandemic is probably too restricted. However, it comes without surprise. Considering the contemporary human society as something in a general pandemic stage, as we will argue, may just be too perplexing.

To focus the picture, first, we must consider that, although this health pandemic has hit all XXI century societies, the etymological meaning of the word is much more comprehensive than being only about health or disease. For instance, the notion pandemic has been applied in biology to indicate species that can be found all over the world; in the very first place, humans included. Subsequently, human practices—be it processes driving climate changes, natural resources exploitation, or economic globalization—also fall into a pandemic definition. They shape the planetary social-ecological systems and, hence, concern all people by affecting all the globe [29]. The examples, climate changes, natural resources exploitation or economic globalization, and other global interactions with the Earth system belong to this broader notion of 'pandemic'. Thus, we must keep in mind that the notion 'pandemic' englobes other events than a rampant disease. Furthermore, subjects of the habitual application space of geoethics belong to the events that fall under the broader connotation of a pandemic.

To show how we like to relate the field of geoethics philosophically with the crisis that a pandemic pose, let us recall a part of the axiomatic definition of geoethics: *"Geoethics consists of research and reflection on the values which underpin appropriate behaviors and practices, wherever human activities interact with the Earth system"* [30] (p. 2). As mentioned before, 'wicked', that is complex-adaptive, social-ecological-systems exhibit a plethora of non-linear characteristics, one of them, the feedback processes involving cultural responses to natural processes. In that sense, the above definition of geoethics might also be read as *"wherever the Earth System and human practices interact"*. This statement expresses the critical property of the Earth system in times of anthropogenic global change, the robust feedback between nature and social components [31]. They form a loop of cause-effect in which who was the cause and what was the effect does make little sense to be determined.

### 2.2. A Word of Caution and Opportunity

We acknowledge that we should be cautious about advancing broad conclusions regarding subjects far away from our field of expertise. We like to avoid falling into Epistemic Trespassing, nor do we intend to give any lesson from our field [32]. We like to enrich the discussion from an interdisciplinary/transdisciplinary point of view, which is a constructive way to do sciences cooperation by sharing experiences. It was not easy, as we usually do (referencing, for example, scientific papers or books), to find respected scholars holding worldwide recognition who did publish academic works to

support our essay of the relation of geoethics with the current health pandemic. In that context, we acknowledge the related work about the broader meaning of the notion Anthropocene, that is, going beyond its geoscience focus and detailing how pandemic are human practices [33].

However, many acknowledgeable contemporary intellectuals recently gave interviews in prestigious publications and media that seem relevant for our present work. If we restrict our sample to some scholars of different generations and diverse schools of thinking, there are even commonalities we cannot disregard. We could cite Alain Badiou, Alain Touraine, Bernard-Henri Levy, Byung-Chul Han, Daniel Innerarity, Edgard Morin, Giorgio Agamben, Jean-Luc Nancy, Jeffry Sachs, John Gray, Judith Butler, Jürgen Renn, Juval Noah Harari, Mariana Mazzucato, Massimo Ferraris, Massimo Cacciari, Noam Chomsky, Paul Krugman, Roberto Esposito, Slavoj Žižek, Svenja Flasspöhler, Thomas Piketty, and many others. Despite their diverse backgrounds, generations, and school of thinking, they all agree with different words that the times following the current pandemic will be a period offering the opportunity to rework the planetary systems.

This renewed 'planetary normal' must be built in a way, for example, to diminish inequalities and increase justice, to protect better the environment, and to fight against poverty that will be higher in the aftermath. The future 'planetary normal' should be more resource-effective, more knowledge-based, and should use less patronizing approaches [34]. In the end, we must create more *"appropriate behaviors and practices wherever human activities interact with the Earth system"* (see above definition of geoethics). Thus, these future actions also may come under the geoethics umbrella, simply.

## 2.3. Thinking Geoethically—Design

Recent works in geoethics did sketch some ethical preferences and the conditions necessary to appropriately 'interact with the Earth system', not just as Earth's natural and social scientists but also as citizens. Among the philosophical tools that geoethics offers, such as emphasis on geo-education or geo-heritage [35], the 'geoethical promise' (Table 1) is strongly linked with moral guidance. The International Association for Promoting Geoethics launched it already in 2014, a blend between a deontological declaration and subtle ethics imperatives.

**Table 1.** Statements at the geoethical promise (Matteucci et al. [5]).

| Statements Made in the Geoethical Promise |
| --- |
| … I will practice geosciences being fully aware of the societal implications, and I will do my best for the protection of the Earth system for the benefit of humankind. |
| … I understand my responsibilities towards society, future generations, and the Earth for sustainable development. |
| … I will put the interest of society foremost in my work. |
| … I will never misuse my geoscience knowledge, resisting constraint, or coercion. |
| … I will always be ready to provide my professional assistance when needed, and I will be impartial in making my expertise available to decision-makers. |
| … I will continue the lifelong development of my geoscientific knowledge. |
| … I will always maintain intellectual honesty in my work, being aware of the limits of my competencies and skills. |
| … I will act to foster progress in the geosciences, the sharing of geoscientific knowledge, and the dissemination of the geoethical approach. |
| … I will always be fully respectful of Earth processes in my work as a geoscientist. |

However, when thinking about probable stages and futures of the Earth System, the World, it deems necessary to go beyond the mainly intra-disciplinary setting of geoethics as expressed, for example, in the geoethical promise. Extended intra-disciplinary frames, like the Cape Town Statement on Geoethics, are necessary and advantageous frameworks, unquestionable [6]. They provide a solid foundation for inter-disciplinary and extra-disciplinary settings of professional geoscientists. However, geoethics should be strengthened for the possible use by any citizen.

Strengthening geoethics may be a rational choice in the Holocene, although it is a must in times of anthropogenic global change. In these times, citizens' actions should *"be judged . . . where they fall on a scale of care and neglect"* because *"[w]hen humans formed an independent relation with the Earth, we were left to choose between a path of care and a path of neglect."* [36] (p. 150). Adequate philosophical tools are required to face such a claim, which enables the human agent to take a *'path of care'* as deems appropriate.

Geoethics was designed for geoscientists as a specific intra-disciplinary tool to choose between care and neglect when studying the interaction of human practices and Earth [14]. A more generic tool should assimilate the ethical foundations that geoethics provide for geo-professionals and simply extend them. Thus, philosophical concepts are needed that focus on the human agent, knowledge, and appropriate behavior and do not bring additional concepts into the design of geoethics, such as, for example, of a contract between humans and nature [37]. The works of Kohlberg [26] and Jonas [38], namely about moral adequacy of normative frameworks of human agents and the imperative of responsibility for those agents who deploy technologies, provide such concepts. The extension of the foundations of geoethics can be done in two steps to describe what *"appropriate behaviors and practices, wherever human activities interact with the Earth system"* [30] (p. 2) implies.

In the first place, the conditions to act appropriately can be linked with the Kohlberg scale of Moral Adequacy (Table 2), in which human behavior shows different ethical responses with increasing quality, according to three levels, divided into two-stage each. At the Pre-Conventional level, human acts by fear of punishment or egoistic convenience. At the Conventional Level, the human behaves following the opinions of the majority of individuals forming a herd, or just because they are educated to follow the laws without questioning them or the authorities. At the Post-Conventional level, people become conscious of their consensus to follow a Social Contract, which equilibrates individual aspirations with the common ones, and the interaction of people is for the greater, common good.

**Table 2.** Kohlberg stages of moral adequacy (adapted).

| Level | Stage | Social Driver |
|---|---|---|
| Pre-Conventional (actions are judged by their direct consequences) | 1 | Obedience and punishment Blind egoism |
| | 2 | Self-interest orientation Individualism, Instrumental egoism |
| Conventional (actions are judged by comparing them to society's views and expectations) | 3 | Interpersonal accord and conformity Others approval, Social relationships |
| | 4 | Law and order Blind compliance, Social systems |
| Post-Conventional (individuals' taking precedence over society's principles; inclusion of basic human rights such as life, liberty, and justice) | 5 | Social contract orientation Agrees on common regulations |
| | 6 | Universal ethical principles Principled self-conscience and mutual respect |

In the second place, the conditions to act appropriately can be linked with Jonas' ethic of responsibility. His philosophy significantly influenced environmental thinking and policymaking. It gains critical importance in times of anthropogenic global change, the new reality for what he saw the emergent possibility

> *" . . . technology apart from its objective works, assumes ethical significance by the central place it occupies in human purpose. Its cumulative creation, the expanding artificial environment, continuously reinforces the particular powers in man that created it, by compelling their unceasing inventive employment in its management and further advance, and by rewarding them with additional success—which only adds to the relentless claim. This positive feedback of functional necessity and reward . . . "*. [39] (p. 178)

Starting from moral theory, Jonas derives an approach on how to deal responsibly with technologies. He argues that the amplified prowess of human practices requires a dedicated philosophy of responsibility acting at the long-range relations between people, in terms both of time and of space,

including far-reaching impacts that might be irreversible. This consideration leads to Jonas' imperative of responsibility for future-oriented ethics that guide actions to be compatible with the durability of human life and the living conditions of future generations.

At the time of shaping them, Jonas looked at the impact of emerging biotechnologies on the living world. Several decades later, the change of the global physical environment shows further impacts of technology and pandemic human practices; adding to the application scope of Jonas' works.

Humans, which now number to more than seven billion and by the end of the century, hence soon, will be eleven billion people, require a globalized society for provision of food, goods, and security to organize a decent life on Earth for them. There is little alternative to it. Resources must be mobilized at a planetary scale to serve all humans. Under such circumstances, globalization is not a question of whether. It is a profound question of how. The manner how the production of food and goods, the use of commodities and natural resources is done is a concern for all people. That is, the technological means and societal forms of production are pandemics in many senses. In that circumstance, Jonas' philosophy calls upon those who shape technologies or, *mutatis mutandis*, who understand the embedding of these technologies in the functioning of the Earth System.

The exponential growth of the number of human people of the last two centuries, which happened like the spread of a virus, has wiped any alternative than to operate at a planetary scale, that is: operating pandemically [40,41]. Unhappily, that period of our recent past did leave us, humankind, also with an unpleasant common heritage, the ongoing anthropogenic global change. The manner how the production of food and goods, the use of commodities and natural resources was organized in the recent past led to the given anthropogenic global change. The current generations of affluent societies bear the primary responsibility for having developed and deployed the technology that made the Anthropocene [42]. Climate change is only the single best-known example. Anthropogenic global change, like climate change, concerns all people; hence, it is a pandemic. The geoscience example at the time-horizon likely is the global sea-level rise that will concern all people [43,44]; hence, is pandemic. In these circumstances, Jonas' imperative of responsibility that initially was conceived to guide technological innovation is the call of duty for inter-generational justice—deeply rooted in geoethics.

Combining approaches of Kohlberg and Jonas with the geoethics approach, a 'geoethical rationale' emerge consisting of six categories (normative preferences) that mutually support each other [7]. As will be sketched below, the geoethical rationale may help as guidance for proper sense-making, decision-making, and action in complex-adaptive social-ecological systems, useful also in the context of the COVID-19 pandemic. In short, the geoethical rationale promotes six normative preferences (categories) to human practices to be, namely: agent-centric, virtue-ethics focused, responsibility focused, knowledge-based, all-agent inclusive, and universal-rights based (Table 3).

The first to fourth category belong to the habitual foundation of geoethics, as formulated so far. Within this group, drawing on Jonas' work, the third category, 'responsibility focused' mentions explicitly future generations. The fourth category, 'knowledge-based' is tuned so that scientific knowledge other than geosciences, is mentioned as well as experience-based knowledge. The general emphasis of a core feature of the scientific methods, namely reproducibility of findings, is mad explicit for an application case beyond sciences. The matters that the fifth and sixth category address could be implicit to geoethics. However, Kohlberg's and Jonas' works render them explicit, and in that manner broaden the application scope of geoethics.

The six categories are not ranked. Each of them is applicable in given social-ecological circumstances; although, their mix and relative weight will vary. Together, they are guiding and challenging the human agent, be it an individual agent, a collective agent, a corporation, or an institutional agent. As their mix and relative weight varies with the circumstances, the geoethical rationale is a means of empowerment. In turn, that feature is paramount for a broadened application scope of geoethics [28].

**Table 3.** Description of the normative preferences of the geoethical rationale; adapted from Bohle [7].

| Normative Preference | General Description |
| --- | --- |
| Agent-centric | To apply a normative framework that invests (empowerment) an individual /group/institution to act to their best understanding in the face of given circumstances, opportunities, and purposes; |
| Virtue-ethics focused | A corpus of personal traits (honesty, integrity, transparency, reliability, or spirit of sharing, cooperation, reciprocity) of an individual/group/institution that furthers operational (handling of things) and social (handling of people) capabilities of the individual/group/institution; |
| Responsibility focused | The outcome of a normative call (internal, external) upon an individual/group/institution that frames decisions/acts in terms of accountability, as well for the intended effects as for unintended consequences and implications for future generations; |
| Knowledge-based | In the first and foremost instance, (geosciences/Earth system) knowledge acquired by scientific methods; experience-based ('indigenous/traditional') knowledge is a secondary instance; reproducibility of knowledge by third parties supports any claim of trustworthiness instead of allusion to faith or 'authorities'; |
| All-agent inclusive | Achieve a practice of a 'shared social license to operate' between various individuals/groups/institutions by mitigating differentials of power or voice; using participatory processes and capacity building; |
| Universal-rights based | Guide effective and rational sense-making of individuals/groups/institutions by universal rights (life, liberty, justice) to strengthen secondary normative constructs such as utilitarian, sustainability or precautionary principles; |

## 2.4. An Upcoming Research Question

During the COVID-19 Pandemic, we have started to wonder whether the Kohlberg scale of moral adequacy may describe what the modus operandi of public governance is, as well as the success and failures of societies in different regions of the world. Many philosophers and intellectuals, whom we listed above (and likely many others), seem to suspect something of that kind.

Although perceived as a kind of aspirational utopia, the higher stage of moral adequacy is the one where people act based on deeply held ethical principles. We can find individuals or some small groups of them acting at that level.

We have accepted that we do not have the proper tools and the capacity to identify societies or countries acting at a specific level of moral adequacy. Doing that would be inappropriate, considering our backgrounds, and a not responsible epistemic trespassing [32]; the test of such assumptions can only be done in a multidisciplinary way, which we hope to trigger in some way.

The COVID-19 pandemic is the first pandemic in a globalized world. It may be a test-ground for assumptions that relate the societal level of moral adequacy to the modus operandi of governance, as well as the success and failures in handling the pandemic in such multidisciplinary way. For example, one could analyze the deaths related to SARS-COV-2 versus 100,000 inhabitants per country given indexes of institutional quality and educational levels, adding other indicators as economic conditions, and the so, traditionally developed by social sciences.

So far, we have some incidental insights of failures in societies that operate merely at the pre-conventional level (fear and social repression). Besides, we see mixed results in societies that seem to combine management according to the pre-conventional (cyber controls, exaggerated individualism) and conventional levels (others approval, conformity). Likewise, we seem to notice confusing results in societies organized to function at the conventional level (law and order, blind compliance). In societies functioning at the post-conventional level, we seem to observe better results (social contract oriented, standardized regulations); maybe because these societies are better organized at governance, knowledge (scientific, technologic, and cultural), economic, and social levels. We are inclined to speculate that for them, the individual interests are better equilibrated with the social ones, helping a better response to a crisis such as the present one, thus having created a better resilient social-ecological system.

At the time of writing this essay, it is too early, with the rampant COVID-19 pandemic fully active, to jump to premature conclusions of such relations, embarking on conclusive studies with the disease still taking thousands of lives. Nevertheless, within such necessary studies, the normative framework

of geoethics may provide a useful analytical framework to support, with the help of other disciplines in a more cooperative way, better practices now and in the future.

The assumption could be formulated preliminarily as "The degree of moral adequacy of societies is one of the most relevant metrics to assess the potential success in fighting disasters affecting the socio-ecological systems. Geoethics offers specific metrics to qualify adequacy".

## 3. Results and Discussion

*Geoethics for Pandemic Contexts*

The application scope of the geoethical promise is mainly intra-discipline with a focus on the field of education in geosciences. The application scope of the geoethical rationale is broad. It is driven by the desire to capture the human condition on Earth in times of anthropogenic global change. Both approaches may be challenged because they side-line other philosophical concepts [10,11,45,46] that are suitable to face circumstances of a pandemic. They may be seen as an idealist-based secular approach making abstractions from the social organization, capitalist determinations, or historical developments, which are necessary to understand pandemics. Regarding the last challenge, the definition of geoethics also stresses the *"the social role and responsibility of geoscientists in conducting their activities"* [30] (p. 2), which provides a foothold to counter such weaknesses, which are paramount in times of a (health) pandemic that at its roots is about social organization. For example, how the 'coronavirus' emerged (markets), how it did spread through societies (travel) causing the illness COVID-19 to be pandemic, or what works to confine the outbreaks (spatial distancing) - any of these courses is mainly about social organization.

In the light of the above, the wording of 'Cape Town Statement on Geoethics' [6] illustrates that reflections on matters like social organization, capitalist determinations, or historical developments are currently nearly absent in geoethics. Instead, the statement is asserting in an aspirational manner that

> "*It is essential to enrich the roles and responsibilities of geoscientists towards communities and the environments in which they dwell . . . Geoscientists have know-how that is essential to orientate societies towards more sustainable practices in our conscious interactions with the Earth system . . . . Geoscientists are primarily at the service of society. This is the deeper purpose of their activity*". (p. 6)

Still, these aspiring insights are incentives to explore the societal contexts of the geosciences [47].

At first sight, technology and practices for deployment determine how people alter natural environments [48,49]. However, economic, and social conditions, cultural constraints, including hegemonic values as well as scientific insights, and financial resources play essential roles. Together these features of a social organization determine which 'endeavors' of anthropogenic change are possible or desirable to undertake. The naïf human endeavor in contemporary times is to operate a 'technosphere' at the planetary scale, which is the essence of the pandemic nature of the present-day human society (see Jonas). Considering these features of contemporary societies, we are confident that the geoethical rationale could apply from Earth Scientists to Epidemiologists, from Health-care personnel to law-enforcement corps, and all the society. Although the geoethical rationale set down its roots into geosciences, that in turn are paramount for the functioning of modern societies, the geoethical rationale is reasonably generic to apply beyond a geoscience context.

We have little doubts that the crisis of the COVID-19 pandemic will alter social organization in many ways. How this will happen and what vision should be found to inspire the necessary changes, appropriate philosophical foundations to "*choose between a path of care and a path of neglect*" (see above) should be required. We direct our efforts to enhance education (from primary to higher, at all levels) [50], which is a way to advance along the Kohlberg scale. Currently, scientific rationalism has been struggling to face the irresponsibility of small leaders. They are managing big problems with fake news, social anxiety and disruption, and conspiracy theories. They are taking decisions denying facts that could interfere with a predetermined political agenda, rejecting evidence-based policies in favor of policy-based 'evidences'.

On the other hand, interestingly, it seems that science is gaining new respect in many sectors of society, despite accusations of some alleged 'arrogance of science' from those not familiar with how sciences work [51]. We must show citizens that science exists and is relevant to them. We shall be affirmative although, we have doubts; although, our 'discoveries' are subject to changes; although, discoveries are not dogmas; although contradictions are part of the scientific work; although it contains uncertainties; and controversies reigns. We do not expect nor look for scientific unanimity in 'regular times' and even less in the present not-so-normal times. Citizens and stakeholders need to be informed about the dialectic nature of the scientific process that would and will appear during this challenging trip to new human knowledge and practices [52,53].

In these circumstances, the call of duty to Earth Scientists is specific. Within society's corpus of technological means, social organization, cultural views, and scientific insights, geoscience knowledge has the potential to fundamentally shape the direction, effectiveness, and efficiency of anthropogenic change of Earth system dynamics [54,55]. To that end, when answering questions about the Earth system like 'where to situate humankind', 'how to change processes', or 'what features to safeguard', the geosciences provide 'instruments'. For example, such instruments are Earth science literacy, insights into the origin of Earth including its development through aeons and understanding how Earth system dynamics operate, and geoethics to guide about the 'ought to be'. When considering the anthropogenic global change in its daily societal context, people need geoscience knowledge because any given individual interacts with the Earth system, be it only as a consumer of resources. Furthermore, citizens need insights into the functioning of the Earth system to engage in better-informed decision-making. Hence, a dedicated responsibility for geoscientists results from the specific function that they have within contemporary societies because of the corpus of expertise that they can offer. To summarize, geoscience expertise is an instrument that made possible the making of global anthropogenic change; that is, making the anthropogenic change pandemic. Geoscientists are co-architects of pandemic anthropogenic global change, for the good and the evil. Therefore, they should assume the responsibility that comes with their role as agents of technology-driven change. In this context, how geoscientists use their expertise is not an impartial matter [16]. They are called to duty to offer cures in the pandemic of global anthropogenic change; that is the essence of geoethics.

Finally, we like to suppose that the managing of the current pandemic in different countries, with diverse governance systems and socio-cultural backgrounds, up to now has shown more efficiency and better results according to the position at the Kohlberg scale of moral adequacy that they hold. We like to suppose further that such observation is linked with the quality of education systems, social and cultural cohesion, and practices of participatory governance. These features also are shown by the quality of political leadership. That linkage seems to be probable because education (including social respect regarding science) is an investment into benefits for future times and generations. The insights into the management of the COVID-19 pandemic [56] seem to teach us that good governance, supported by the best scientific knowledge, and applying the higher ethical standards, is the one showing the best results. In that context, we would like to propose that geoethics, if rethinking its normative preferences in terms of institutional or collective agents, is about governance practices appropriate for pandemic circumstances, which, in turn, are the essence of the human condition in times of anthropogenic global change.

## 4. Pandemical Geoethics?

Nudging by moral imperatives is crucial at the present historical moment that is unique for our societies, although it is less than unique at a historical scale. Notwithstanding the latter, the feeble historical memory in our societies makes the COVID-19 health pandemic a unique experience for most people. However, as the experiences are not unique on historical scales, we can refer to imperatives that root in our cultures. Only a few of them need to be mentioned to illustrate our choice that was outlined above.

The following shortlist does present choices within our (limited, western) cultural frames on how to cope with the COVID-19 crisis:

- Act as if the maxims of your action were to become through your will a universal law of nature. (I. Kant)
- Act in such a way that you treat humanity, whether as your person or as the person of any other, never merely as a means to an end, but always at the same time as an end. (I. Kant)
- Act so that the effects of your action are compatible with the permanence of genuine human life. (H. Jonas)
- Enjoy life and help live. (M. Bunge)

The first two are Kant's Categorical Imperative [57]. It is here formulated in two sentences and can be linked to the universal Golden Rule presents in Western and Eastern cultures from ancient times ('Do not impose on others what you do not wish for yourself'). As the pandemic affects the entire planet, the maxims guiding human actions should be considered universal laws, and all the people must be treated as means and ends, equally.

When, as comes clear in hindcast, geoethics internalize Kant's imperatives, it is also pushing Earth scientists and citizens to answer Kant's questions [21] *"What can I know? What should I do? What may I hope?"* when confronted with interactions with the Earth system. All these questions are embedded into the origin of the Geoethical Promise (see Table 1). When COVID-19 began to spread, the duty of all, scientists, decision-makers, and citizens was to know what could be known in these given circumstances. Then, with some level of knowledge, the definition of what should be done would be the second step to finally provide quality scenarios to decision-makers and citizens. Probably, most of the initial failure to cope with the pandemic was the little we understood and the uncertainty regarding proper reactions. When limiting our actions to Kant's imperative, the problem is that we shall care only about the goodness of our actions, despite the result of them. The failures of actions by decision-makers during the COVID-19 crisis that were taken with good intentions show that following the rule of the Categorical Imperative is insufficient for satisfactory results. Max Weber mentions in his essay 'Politics as a Vocation' (*Politik als Beruf, 1919*) [22,23] that a politician (or a decision-maker) needs to balance an 'Ethic of Moral Conviction' that defines duties, with an 'Ethic of Responsibility', which oversees the consequences.

Correspondingly, geoethics is evolving beyond a deontological moral theory (where the rightness or wrongness of actions does not depend on their consequences but on whether they fulfill a duty). Geoethics offers a concept of ethics in Earth sciences where other imperatives enrich and sustain better human interactions with the Earth system. Thus, we can add that to confront and overcome a pandemic we, geoscientists who also are citizens need to follow the Categorical Imperative as well as to answer the three Kantian questions adaptively and act based on other few complementary imperatives.

Using the biomedical progress of knowledge as an example, at 'The Heuristic of Fear', Jonas [58] appeals to a prudent responsibility. He presented an imperative that is deduced by a 'heuristics' concept, which is a practical method for problem-solving, not perfect nor fully rational but allowing attending short-term goals. Prudent responsibility is mandatory when predictions of a negative outcome are more likely than predictions of a good outcome. Prudent responsibility is a reliable alternative to guide the actions when complexity and uncertainties prevail, as in complex-adaptive social-ecological systems. Prudent responsibility is an upgrade of an ethic of the 'Precautionary Principle', which demands that uncertainty must not be a reason for inaction [59]. During the COVID-19 pandemic, we saw diverse ways of managing the fight against the disease, conflicting scientific results, and, at the decision-making level, the struggle between the health and economy. Following Jonas' ideas, we can identify this as a false dilemma because our predictions from both fields are not good enough to make an entirely rational decision. Thus, parsimony must prevail, and heuristic approaches are useful tools (e.g., trial and error, rule of thumb, educated guess). This heuristic approach also fits Bunge's systemic thesis of truth about how to manage the non-reproducibility of many scientific

findings [60]. Likewise, a heuristic approach matches the feature of irreversible path-dependency that many complex-adaptive social-ecological systems exhibit as well as Weber's distrust in monocausal explanations because in complex systems there can be multiple causes producing the same effect. In Geosciences, the Precautionary Principle is a practical ethical instrument used in many cases, from natural or human-made hazards and risks to natural disaster mitigation and social-ecological adaptation. The Pacific Tsunami Warning system, for instance, alerts the coastal states of the potential generation of tsunamis. However, even with a significant rate of false alarms, the prudent management of evacuations prevails. In such sense, and given the poor scientific knowledge regarding COVID-19 we held, precautionary actions prevailed in most of the cases, based not on strong scientific rationalism but in heuristics approaches.

Geoethics, combining approaches of Kohlberg (Table 2) and Jonas' imperative [1,24], advocates human agents to target the highest levels of Kohlberg scale of moral adequacy. Furthermore, it advocates following the imperative of responsibility for those agents who deploy technologies, not just for the present society and planet but also considering the future generations to come. Jonas proposed that we must review ethical theory to incorporate the concept of "responsibility" into the center of ethics, considering that duties, solely, are not enough for such intergenerational obligation. In other words, ethical reflections must be upgraded accordingly with technological advancement and the future consequences and responsibilities of current human actions. Geoethics is following that path by examining innovative ideas of ethical theory and investigating into the nature of responsibility. This responsibility goes from top government officials to lay citizens. At the moment of publishing this paper, the lack of responsibility seems to be behind many new outbreaks. The geoethical rationale and its six normative preferences (agent-centric, virtue-ethics focused, responsibility focused, knowledge-based, all-agent inclusive, and universal-rights based) sketch ethical and societal actions. They may apply not just during the COVID-19 pandemic, but also for future generations, by offering a legacy of responsible actions.

Finally, we may examine the COVID-19 crisis under the light of Bunge imperative [61], pushing us into the search for happiness as a fundamental universal human right, but not as a gift and without burdens, but intimately linked with the duty of helping others lives. Thus, to enjoy life during the pandemic, we must take care of our health and we must help others to live (by special distancing, quarantine if necessary, and any other measures heuristically or scientifically suggested by universal insight or the authorities). That means we must find the proper balance between our rights and our social duties. Individual adherence to these social impositions, conventions, or social contracts will indicate on which Kohlberg stage we are, at least to ourselves.

On these imperatives or moral maxims, we have much, from the ethical point of view, to rework the complex-adaptive social-ecological system of planet Earth during and, mostly, in the aftermath of the given pandemic. Our plea is for imperatives for the good Earth governance, based on the best scientific knowledge, and deeply rooted in ethical values. Kant's imperative established duties, Jonas' imperative defined responsibilities, and Bunge's one, finally, allows us the right to enjoy life or to search for happiness, while imposing the duty to help others enjoy it.

The consequences of suchlike imperatives drive obligations of the present generation towards the future ones, not just the already born now transiting the childhood but also the long sequence of unborn generations to come. We have the ethical obligation of leaving a heritage to humankind offering at least equal, ideally better, opportunities to enjoy life than the ones we had when we arrived in this world. To do so, we need to find the right equilibrium among our rights (and desires) for happiness and our duties, being responsible as individuals and as members of our communities and environment (the social-ecological system) to assure to future generations at least the same rights to enjoy life too.

*A Preliminary Summary: Ad Dictum Futures*

The current turmoil of the early year 2020, which is caused by the COVID-19 pandemic, stretches our imagination to the breaking point. Therefore, irritation rules and ruthless action is observed [62]. Under such circumstances, a metaphorical closure of the given debate about geoethics in times of pandemics seems suitable; three perspectives are sketched below.

First; for many of our fellow citizens the Future, with capital 'F', is the march towards a place called 'about-the-same'. It may be a bit more of the same. For most people, 'the Future' is nothing that is made. It is something to be endured. Moreover, disasters or wars deem ready to disrupt its regular gait. It is this aeon-old view *'nihil sub sole novum'* (nothing new under the sun) that for many provides a sense of security. Astonishingly, 'the Future' is a reference frame. It embeds our myopic starring at the next turn of events. However, what to do, when this reference frame seems to change, to wobble and, hence gets uncertain. Then, menacingly, 'The Unknown' frames the stages of our plays. Irritatingly, 'The Counter-Intuitive' seems to consolidate out of our plays. Threateningly, they block the way back. The horsemen of the modern apocalypse, 'The New', 'The Unknown', 'The Fear', and 'The Counter-Intuitive' threat with insecurity, loss of competences, altered divisions of societies, and lost sense! We were strapped out of our comfort zone by the horsemen. How can we find a new comfort outside our old comfort zone, as we are moving into a 'new normal'? Are we those optimistic and fearless of leaving our comfort zones (because outside them are new universes, duties, responsibilities, and, probably, new happiness)? Or are we the pessimistic ones, fearing 'The New', 'The Unknown', and 'The Counter-Intuitive', preferring to keep ourselves anchored inside our allegedly warm and paradisiac old comfort zone?

Second; some people relish the 'The New', 'The Unknown', and 'The Counter-Intuitive' overcoming 'The Fear'. Artists, explorers, creators, scientists feel a deep sensual pleasure when confronting them, as persons and as citizens. The artist's psyche, the explorer's spirits, the entrepreneur's energy, the innovator's minds, the researcher's souls are resources vibrating with imagination and passion. Hence, nurtured by them, the citizenries may confront 'The Fear' of a future of Quantum-Technology, Earth System Sciences, Advanced Health Care systems, Artificial Intelligence, and Synthetic Biology. Then the citizenries can draft the guides to these galaxies. They will tell, whether '42' is still the right answer, why your towel might be a reliable resource, and who, finally, moved the restaurant(s) at the end of the universe(s)? (references to metaphors used in the trilogy "Hitchhiker's Guide to the Galaxy" by Douglas Adams).

Third; only when behaving as citizens, then the artists, cultural practitioners, inventors, entrepreneurs, and scientists can push the boundaries of the human imagination. Only as citizens, jointly they may move beyond the familiar and transcend the borders towards the Future. Nevertheless, are they ready to assume this task? Do they invest collaboratively in path-changing discoveries, different fates of our planet, and charting pathways to liveable futures? Only then, 'The New', 'The Unknown', 'The Fear', and 'The Counter-Intuitive' will face the broad, vigorous smile of 'The Imaginator'—Surrender! [63]

**Author Contributions:** Conceptualization, E.M. and M.B.; methodology, E.M. and M.B.; writing—original draft, E.M. and M.B.; writing—review and editing, E.M. and M.B. All authors have read and agreed to the published version of the manuscript.

**Funding:** This research received no external funding.

**Acknowledgments:** We would like to thank the reviewers.

**Conflicts of Interest:** The authors declare no conflict of interest.

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
