# Peer review of "Geoethics for Nudging Human Practices in Times of Pandemics"

_sustainability, doi:10.3390/su12187271_

Round 1

Reviewer 1 Report

The article is an innovative and relevant reflection that I considered worthy of publication.

However, the theoretical article is more about Ethics, Sustainability, and Pandemics (with many references to climate change) than really about Geoethics and Pandemics. The authors clarify very well the "Geoethics Promise" and the" Cape Town Statement", but they should better present "the how and why" Geoethics is important in pandemics times? How does the pandemics affect the "Planetary system"?   Why and how  Geoethics values and principles can lead citizens to evolve to the top stage of the Kolbergs scale?

I give two suggestions to authors: (1) change the title to one more focused in the content of the article, or 2) improve the article explaining how and why Geoethics is important to Humans practices in pandemics times ( specify why Geothics contributions are unique in pandemics times) 

Author Response

We like to thank the three reviewers (R1, R2, and R3) for the analyses of our essay, as well as their respective comments, and the various perspectives that they offer. We learned from the remarks of the reviewers to guide better the expectations of readers adjusting the essay accordingly, we hope.

In the attached file, we respond per subjects and in the mutual context of the remarks of the three reviewers. We provide a single balanced response letter to all reviewers to give them the full perspective of the reasons for the introduced changes and improvements but referring to each one at each particular case on several occasions.

Reviewer 2 Report

The following improvements should be considered.

  1. The introduction section needs significant improvement. Half of the introductory section is obsolete. For instance, the paper starts very wide, but very little is said: "Earth scientists did develop the field of geoethics with the initial aim of establishing necessary 48 deontological codes. Very soon, the initial concepts were expanded, updated and reformulated [2]. 49 New ontological perspectives and epistemological analysis were added. Furthermore, at least 50 initially and provisionally, some axioms were established [3]. 51
    A new knowledge began to be built, and the group of scholars dedicated to this field was and is 52 expanding, incorporating colleagues from the areas of philosophy and the social sciences. The planet 53 Earth stopped being seen overwhelmingly from the natural sciences; perspectives of culture, 54 communication and sense-making emerged as part of a more comprehensive picture of the human 55 condition [4]. Likewise, anthropocentric visions of submitting nature to human practices, like 56 ecomodernism, are eluded."
  2. The authors should emphasize the background of the research, significance of the study, and the main framework used. Furthermore, the methods used should be explained. Better connection with the COVID pandemic should be established in the initial section.
  3. There is also a need to mention clearly if there was a call for research and gap exists in the current literature which is the need of the time in terms of the current study to create a strong argument and justification about the contribution.
  4. Methods are not clearly presented and do not allow replication of the study. My main concern is the lack of clear use of the analysis framework. Better qualitative analysis and a more objective assessment fitting to the framework is mandatory.
  5. Implications have to be improved: how your results add something new at the scientific literature on this topic? What does this paper add?  It is not clear how the paper can contribute to filling a research gap. State clearly the novelty of your study.

Overall, the paper needs a better structure and coherency. The theoretical framework should be reviewed in the 2nd section, followed by a clearly explained methods and procedures. Results should be analyzed, presented, and discussed with more scientific rigor.

The following improvements should be considered.

  1. The introduction section needs significant improvement. Half of the introductory section is obsolete. For instance, the paper starts very wide, but very little is said: "Earth scientists did develop the field of geoethics with the initial aim of establishing necessary 48 deontological codes. Very soon, the initial concepts were expanded, updated and reformulated [2]. 49 New ontological perspectives and epistemological analysis were added. Furthermore, at least 50 initially and provisionally, some axioms were established [3]. 51
    A new knowledge began to be built, and the group of scholars dedicated to this field was and is 52 expanding, incorporating colleagues from the areas of philosophy and the social sciences. The planet 53 Earth stopped being seen overwhelmingly from the natural sciences; perspectives of culture, 54 communication and sense-making emerged as part of a more comprehensive picture of the human 55 condition [4]. Likewise, anthropocentric visions of submitting nature to human practices, like 56 ecomodernism, are eluded."
  2. The authors should emphasize the background of the research, significance of the study, and the main framework used. Furthermore, the methods used should be explained. Better connection with the COVID pandemic should be established in the initial section.
  3. There is also a need to mention clearly if there was a call for research and gap exists in the current literature which is the need of the time in terms of the current study to create a strong argument and justification about the contribution.
  4. Theoretical framework and literature should be enriched. Specific hypotheses can be included to better establish a relationship with COVID-19. 
  5. Methods are not clearly presented and do not allow replication of the study. My main concern is the lack of clear use of the analysis framework. Better qualitative analysis and a more objective assessment fitting to the framework is mandatory.
  6. Implications have to be improved: how your results add something new at the scientific literature on this topic? What does this paper add?  It is not clear how the paper can contribute to filling a research gap. State clearly the novelty of your study. Better connect the topic with COVID pandemic.

Overall, the paper needs a better structure and coherency. The theoretical framework should be reviewed in the 2nd section, followed by a clearly explained methods and procedures. Results should be analyzed, presented, and discussed with more scientific rigor.

Author Response

We like to thank the three reviewers (R1, R2, and R3) for the analyses of our essay, as well as their respective comments, and the various perspectives that they offer. We learned from the remarks of the reviewers to guide better the expectations of readers adjusting the essay accordingly, we hope.

In the following sections, we respond per subject and in the mutual context of the remarks of the three reviewers. We provide a single balanced response letter to all reviewers to give them the full perspective of the reasons for the introduced changes and improvements but referring to each one at each particular case on several occasions.

Reviewer 3 Report

The paper is interesting, well-structured and opens up a complex discussion on geoethics.

I’m also convinced that geoethics should not be a discipline that is limited only to attributing the responsibilities, obligations and practices of sharing the correct behaviors to be adopted towards the planet, to geoscientists, but also to scholars of human and social sciences they must make a contribution in this direction. I also agree on the need for a bottom up approach necessary for the communication of science, with the involvement of citizens and stakeholders, even if this task is not always easy and it also requires a common, shared and coherent effort by all scientists. Indeed, they must also fight against the existing prejudice of the population towards science and scientists. On this issue, I invite the authors to also cite some empirical data that reinforces the argument: for example, the Eurobarometer surveys on the relationship between population and science and technology.

On the scholars mentioned (Kohlberg, Jonas, Kant, Bunge), certainly their language and their propositions are consistent with the proposed contents and objectives of geoethics, also in the context of the correct behaviors to mitigate the impact of extreme events such as a pandemic.

I really like the connection between the pandemic and other types of disasters related to extreme natural phenomena such as climate change.

However, on the social contract, I advise the authors to refer also to the thought of Michel Serres (Le Contrat naturel, François Bourin, Paris 1990), while on the ethics of responsibility, I also recommend to consider the reflections of the mathematician and semiotician Charles Sanders Peirce in Ethics of Terminology (1903), also mentioned in this paper (research as humanity’s greatest ethical task): DOI: 10.3280/RGI2019-002002.

Lines 292-300: the authors should also cite some bibliographical references on the influence of cultural, educational, social and institutional factors useful in alleviating the effects of an extreme natural event, not only considering the pandemic but also the disasters linked to extreme natural phenomena (earthquakes, floods, landslides etc.)

Line 360: Furthermore, I advise the authors not to use the expression "natural disasters" because, while some hazards are natural and inevitable, the resulting disasters have almost always been caused by human actions and decisions.

Several researchers around the world have created a group according to which the term 'natural disaster' is factually incorrect and misleading (see campaign #NoNaturalDisasters, https://www.nonaturaldisasters.com/), aiming to change the terminology.

General aspects: Begin with a clear problem statement. What is geoethics and what are the shortcomings of this discipline understanding that motivated you to write this paper? Include concise objectives that follow logically from the problem statement. Mention in the introduction how you plan to develop the discourse on geoethics by linking it to the imperatives and propositions of the scholars and philosophers mentioned.

Author Response

(The authors gave the same response as above.)

Round 2

Reviewer 2 Report

The improvements to the paper have been substantial. 

Still some minor language improvements, and adopting a more formal style is advised. Proofreading is suggested.

For instance line 476 "intimately linked with the duty of helping others lives. To enjoy life, we must be healthy enough. Thus, we must take care of our health during the pandemic, especially, and help others live"

Check the manuscript and eliminate or rephrase such and similar parts that are too informal and redundant. Also, authors may consider further supporting the link to COVID-19.

Especially, you can reference recent studies that evaluated COVID-19 in various contexts. These are just some:

Rosenbaum, L. (2020). Facing Covid-19 in Italy—ethics, logistics, and therapeutics on the epidemic’s front line. New England Journal of Medicine, 382(20), 1873-1875.   Sobirova, Z. (2020). Hoarding and Opportunistic Behavior During Covid-19 Pandemics: A Conceptual Model of Non-Ethical Behavior. International Journal of Management Science and Business Administration, 6(4), 22-29.   Obrenovic, B., Du, J., Godinic, D., Tsoy, D., Khan, M. A. S., & Jakhongirov, I. (2020). Sustaining Enterprise Operations and Productivity during the COVID-19 Pandemic:“Enterprise Effectiveness and Sustainability Model”. Sustainability, 12(15), 5981.

Author Response

Dear Reviewer,

Thanks for your suggestions. We have got your advice regarding the English improvements not only on Grammar and Spelling but also in both informal phrases and redundancies. We submitted the text to further review.

We also added in the right places new references as you suggested and others we found also appropriate. Thanks so much for that, because with the high number of papers being published around the COVID issue, the reference mining is a difficult task, and your suggestions helped a lot.

Further links to COVID are also in our plans, in a book we are co-editing, with contributions of scientists from other fields than geosciences.

Thanks a lot

Eduardo Marone